



# Growth of Atmospheric Freshly Nucleated Particles: A Semi-Empirical Molecular Dynamics Study

Yosef Knattrup, Ivo Neefjes, Jakub Kubečka, and Jonas Elm

Department of Chemistry, Aarhus University, Langelandsgade 140, 8000 Aarhus C, Denmark

**Correspondence:** Jonas Elm (jelm@chem.au.dk)

**Abstract.** When simulating new particle formation rates, collisions in the system are approximated as hard spheres without long-range interactions. This simplification may lead to an underestimation of the actual formation rate. In this study, we employ semi-empirical molecular dynamics (SEMD) at the GFN1-xTB level of theory to probe the sticking process of the monomers sulfuric acid (SA), methanesulfonic acid (MSA), nitric acid (NA), formic acid (FA), ammonia (AM), methylamine
(MA), dimethylamine (DMA), and trimethylamine (TMA) onto freshly nucleated particles (FNPs). The FNPs considered are $(SA)_{10}(AM)_{10}$, $(SA)_{10}(MA)_{10}$, $(SA)_{10}(DMA)_{10}$, and $(SA)_{10}(TMA)_{10}$.

In general, we find that the hard-sphere kinetic approximation, which neglects long-range interactions, significantly underestimates the number of collisions leading to sticking. By calculating the sticking coefficient from SEMD simulations, we obtain enhancement factors of 2.3 and 1.5 for the SA+$(SA)_{10}(AM)_{10}$ and AM+$(SA)_{10}(AM)_{10}$ collisions, respectively. A comparison
with OPLS all-atom force field simulations shows similar enhancement factors of 2.4 and 1.6 for the SA+$(SA)_{10}(AM)_{10}$ and AM+$(SA)_{10}(AM)_{10}$ collision, respectively.

Compared to the force field simulations, SEMD exhibits a more isotropic sticking behavior, with the probability remaining near unity for small offsets before rapidly dropping to 0% beyond a certain offset. In contrast, the force field simulations show a more gradual decline in sticking probability due to certain orientations still leading to sticking. The largest discrepancy
between the two methods occur at lower collision velocities—below 200 m/s for SA and below 400 m/s for AM—where force field simulations, even for head-on collisions, predict low or zero sticking probability. This has previously been attributed to periodic repulsions between the rotating collision partners caused by fluctuations in their charge distributions. In contrast, SEMD simulations do not exhibit this behavior. Since these low velocities are not significantly populated in our simulations, both methods yield similar enhancement factors. However, for systems with larger effective masses, where such velocities are
more prevalent, we would expect the two methods to diverge.

## 1 Introduction

According to the 6th assessment report of the IPCC, aerosols are responsible for the largest uncertainty in current global climate models, due to their ill-defined effects on the climate (Lee et al., 2023). Aerosols affect the climate by direct contact with sunlight through absorption or reflection, or by acting as nucleation cores, termed cloud condensation nuclei (CCN), for
water uptake and further growth into clouds (Boucher and Lohmann, 1995). Of these two factors, the uncertainty associated





with the formation of CCN has the greatest impact on modeling radiative forcing, (Canadell et al., 2021) and this uncertainty is mainly due to a lack of understanding of the initial formation steps leading to particles of 1.7–3.0 nm in diameter. For instance, changing the growth mechanism of 1.7–3.0 nm particles from being solely sulfuric acid-based to including extremely low volatility organic compounds can lead to a factor of two difference in the predicted CCN number concentration (Tröstl et al.,
2016). Hence, understanding the growth behavior of freshly nucleated particles, around 1.7 nm in diameter, is paramount for reducing this uncertainty.

Modeling studies show that around 30–60 % of CCN, over land, are formed through new particle formation (NPF), a process where atmospheric gas-phase precursors undergo gas-to-particle conversion (Boucher and Lohmann, 1995; Merikanto et al., 2009; Zhao et al., 2024). This leads to a burst of 1–2 nm freshly nucleated particles (Kulmala et al., 2013). The NPF process
can to a large extent be explained by clustering of sulfuric acid (SA) coupled with bases of high abundances such as ammonia (AM) (Kulmala et al., 2013; Kirkby et al., 2011; Schobesberger et al., 2013; Weber et al., 1996; Elm, 2021a; Dunne et al., 2016) or high basicity such as the alkyl-amines methylamine (MA), dimethylamine (DMA) (Elm et al., 2020; Engsvang et al., 2023b; Kurtén et al., 2008; Loukonen et al., 2010; Nadykto et al., 2011, 2015, 2014; Jen et al., 2014; Glasoe et al., 2015; Almeida et al., 2013; Elm, 2021a) and trimethylamine (TMA) (Elm, 2021a; Kurtén et al., 2008; Nadykto et al., 2011; Jen et al.,
2014; Nadykto et al., 2015; Glasoe et al., 2015). Cluster formation has also been shown to be enhanced by other acids such as nitric acid (NA) (Wang et al., 2020; Liu et al., 2021, 2018; Kumar et al., 2018; Ling et al., 2017; Wang et al., 2022; Nguyen et al., 1997; Longsworth et al., 2023; Knattrup et al., 2023; Knattrup and Elm, 2022; Bready et al., 2022; Qiao et al., 2024), formic acid (FA) (Bready et al., 2022; Knattrup et al., 2023; Ayoubi et al., 2023; Zhang et al., 2022, 2018; Harold et al., 2022; Nadykto and Yu, 2007) or methanesulfonic acid (MSA) (Elm, 2021b; Dawson et al., 2012; Chen et al., 2015, 2016; Perraud
et al., 2020; Arquero et al., 2017b, a; Chen and Finlayson-Pitts, 2017; Elm, 2022).

NPF rates can be modeled using cluster distribution dynamics simulations, where so-called birth–death equations are solved for the relevant formation pathways,

$$\frac{\mathrm{d}c_i}{\mathrm{d}t} = \sum_{i=1}^{\lfloor i/2 \rfloor} s_{j,(i-j)} c_j c_{(i-j)} + \sum_j \gamma_{(i+j)\to i} c_{i+j} - \sum_j s_{i,j} c_i c_j - \sum_j^{\lfloor i/2 \rfloor} \gamma_{i\to j} c_i, \tag{1}$$

where $c_i$ is the concentration of cluster $i$, $t$ the time, $\gamma_{i\to j}$ the evaporation of $i$ to form $j$, and $s_{i,j}$ the sticking coefficient
for sticking collisions between $i$ and $j$. The sticking coefficient $s_{i,j}$ is related to the collision coefficient $\beta_{i,j}$ through an accommodation factor $\alpha_{i,j}$ which measures the percentage of collisions that result in a stable cluster,

$$s_{i,j} = \alpha_{i,j} \beta_{i,j}. \tag{2}$$

The Atmospheric Cluster Dynamics Code (ACDC) constructs the equations and invokes an ODE solver routine to solve the birth–death equations numerically for a given set of cluster sizes (McGrath et al., 2012). ACDC determines evaporation coeffi-
cients based on the binding free energies of the clusters, using detailed mass balance under the assumption that the evaporation rate does not change significantly from equilibrium (Ortega et al., 2012). Binding free energies are typically calculated using quantum chemistry methods, performed by searching for the cluster configuration with the lowest Gibbs free energy. In recent





years, significant research efforts have focused on improving the accuracy of binding free energy calculations for clusters (Elm et al., 2020, 2023; Engsvang et al., 2023b).

Sticking coefficients, on the other hand, are often approximated as collision coefficients calculated through kinetic gas theory, i.e., $\alpha_{i,j} = 1$. In this framework, the collision partners are represented by hard spheres of defined radii, without any long-range interactions between them, meaning that a clustering event only occurs if the hard spheres overlap. By default, ACDC employs this approach to calculate sticking coefficients. In reality, atmospheric molecules and clusters interact through long-range interactions, which can significantly enhance the collision coefficient. Furthermore, it is not clear if the unit accommodation

factor is a reasonable approximation for all clusters of interest in NPF.

Experimentally, Stolzenburg et al. (2020) showed that the growth rates of uncharged SA particles from 1.8 to 10 nm exceed the hard-sphere kinetic limit for the condensation of SA. Similarly, Stolzenburg et al. (2018) studied growth with organics in the 2 to 30 nm range and found that organic condensation drives particle growth. However, measurements of particles at and below 1.7 nm in diameters are lacking.

Recently, several studies have used molecular dynamics (MD) simulations to model sticking coefficients of atmospheric relevant systems. MD simulations track the system's dynamic evolution, directly accounting for dynamical effects such as the explicit temperature, i.e., the thermal motion. This is unlike the statistical thermodynamic treatment employed in quantum chemistry, where results are "extrapolated" to the given temperature from the zero Kelvin structure.

Halonen et al. (2019) used force field molecular dynamics to study collisions between two SA molecules, finding a collision

coefficient 2.2 times larger than that predicted by kinetic gas theory. Yang et al. (2023) investigated collisions of SA and DMA molecules with bisulfate–dimethylammonium clusters containing up to 16 dimers using similar force field methods. They observed that the enhancement over kinetic gas theory decreases with increasing cluster size. Other force field MD studies have explored the formation of small atmospheric clusters of SA with W or bases (Anderson et al., 2008; Loukonen et al., 2014b) or with ions (Neefjes et al., 2022; Halonen et al., 2023).

However, classical force field methods cannot model bond breaking and, therefore, cannot account for chemical reactions, such as proton transfers, which are critical for stabilizing atmospheric acid–base clusters. When acid–base clusters are used as collision partners, as in Yang et al. (2023), the fixed bonds of the monomers fail to accurately represent the interactions within the cluster. Additionally, chemical reactions, such as proton transfers, cannot occur after collisions, meaning the stability and sticking efficiency of the cluster after a collision cannot be accurately assessed.

Loukonen et al. (2014a) used DFT-based MD simulations to investigate collisions of the SA–DMA and SA–DMA–W systems, finding a sticking coefficient of unity due to proton transfer reactions. However, their study was limited to head-on collisions between molecules and dimers, as DFT-based MD simulations are computationally expensive.

Cluster distribution dynamic simulations have generally been limited to relatively small clusters (e.g., 8–10 molecules) due to the increasing computational cost of binding free energy calculations for increasing cluster sizes. Recently, however, much

larger atmospheric-relevant systems, referred to as "Freshly Nucleated Particles" (FNPs), have been studied. Engsvang and Elm (2022); Engsvang et al. (2023a) studied $(SA)_n(AM)_n$ clusters, with $n$ up to 30, while Wu et al. (2023, 2024) studied $(SA)_n(base)_n$ clusters, with $n$ up to 15 and the bases AM, MA, DMA or TMA. Wu et al. (2024) determined the cluster-





to-particle transition point, where clusters begin to exhibit bulk particle-like properties, as occurring at cluster sizes of 8–10 acid–base pairs. These $(SA)_{10}(base)_{10}$ systems, with geometrical diameters reaching up to 1.8 nm, serve as ideal test systems
for investigating the growth of FNPs by collisions with monomers.

In this work, we study the collisions of the FNPs $(SA)_{10}(AM)_{10}$, $(SA)_{10}(MA)_{10}$, $(SA)_{10}(DMA)_{10}$, and $(SA)_{10}(TMA)_{10}$ with the atmospheric precursor vapors SA, MSA, NA, FA, AM, MA, DMA, and TMA in the free molecular regime. The collisions were studied explicitly using Born–Oppenheimer MD at the semi-empirical GFN1-xTB level of theory (Grimme et al., 2017). The GFN1-xTB method extends beyond traditional force field approaches by enabling the modeling of chemical
reactions, such as proton transfers, and accounting for the dynamical charge distribution, while maintaining significantly greater computational efficiency than DFT-based MD. By comparing sticking coefficients derived from these simulations with those predicted by kinetic gas theory and force field MD simulations, we aim to assess the impact of long-range interactions and quantum mechanical effects in modeling the growth of FNP particles.

## 2 Methodology

### 2.1 Computational Details

All MD simulations were carried out using the Atomic Simulation Environment (ASE) (Larsen et al., 2017). At each time step, the forces on the nuclei were calculated using the semi-empirical GFN1-xTB method (Grimme et al., 2017) using the xtb-python calculator (Grimme-Lab, 2022). Between time steps, the nuclei are propagated classically. A time step of 1 fs was used for all simulations, as it was found to be sufficient to resolve the hydrogen stretch vibration. Prior to the collision trajectory
simulations, each monomer and cluster were separately equilibrated to ensure thermal equilibrium.

### 2.2 Equilibration Procedure

Atmospheric molecules and clusters are assumed to be thermally equilibrated. To ensure the studied molecules and clusters are at thermal equilibrium before the collision simulation, we adopted the approach outlined by Halonen et al. (2023).

We equilibrated each molecule or cluster in a separate NVT ensemble simulation, using the Langevin thermostat with a target
temperature of 300 K and a time constant of 100 fs. Initially, atomic velocities were randomly assigned based on the Maxwell–Boltzmann relative velocity distributions at 300 K. The equilibration simulations were run for 10 ns for molecules and 1 ns for clusters. Thermodynamic properties were saved every 500th time step to minimize correlations between consecutive output frames. We used equipartition as the measure for thermal equilibrium, i.e., when the cumulative average of the translational, rotational, and vibrational temperature components aligns with the target. The progression of equilibration was tracked by
monitoring the cumulative averages of these temperature components over time. The total instantaneous temperature $T_{\text{tot}}$ at any given time step is calculated from the kinetic energy $E_{\text{kin}}$ as:

$$T_{\text{tot}} = \frac{2E_{\text{kin}}}{3k_{\text{B}}N},$$
(3)




where $N$ is the number of atoms and $k_B$ the Boltzmann constant. The corresponding translational ($T_{tr}$), rotational ($T_{rot}$), and vibrational ($T_{vib}$) temperature partitions are given as:

$$T_{tr} = N\left(T_{tot} - T_{com}\right), \tag{4}$$


$$T_{rot} = N\left(T_{com} - T_{rotate}\right), \tag{5}$$

$$T_{vib} = \frac{N}{N-2}T_{rotate}, \tag{6}$$

where $T_{com}$ is the temperature after subtracting the center-of-mass (COM) motion, and $T_{rotate}$ the temperature after subtracting both the COM and rotational motions.

### 2.3 Collision Simulations


The collision simulations were carried out using the velocity Verlet algorithm, approximating the atmosphere as a free molecular regime. The simulations were initialized by placing the collision partners at a COM distance of $r = 20$ Å from each other, offset by $b$ and with the initial relative velocity $v$ as illustrated in Figure 1. The placement distance is motivated by assuming

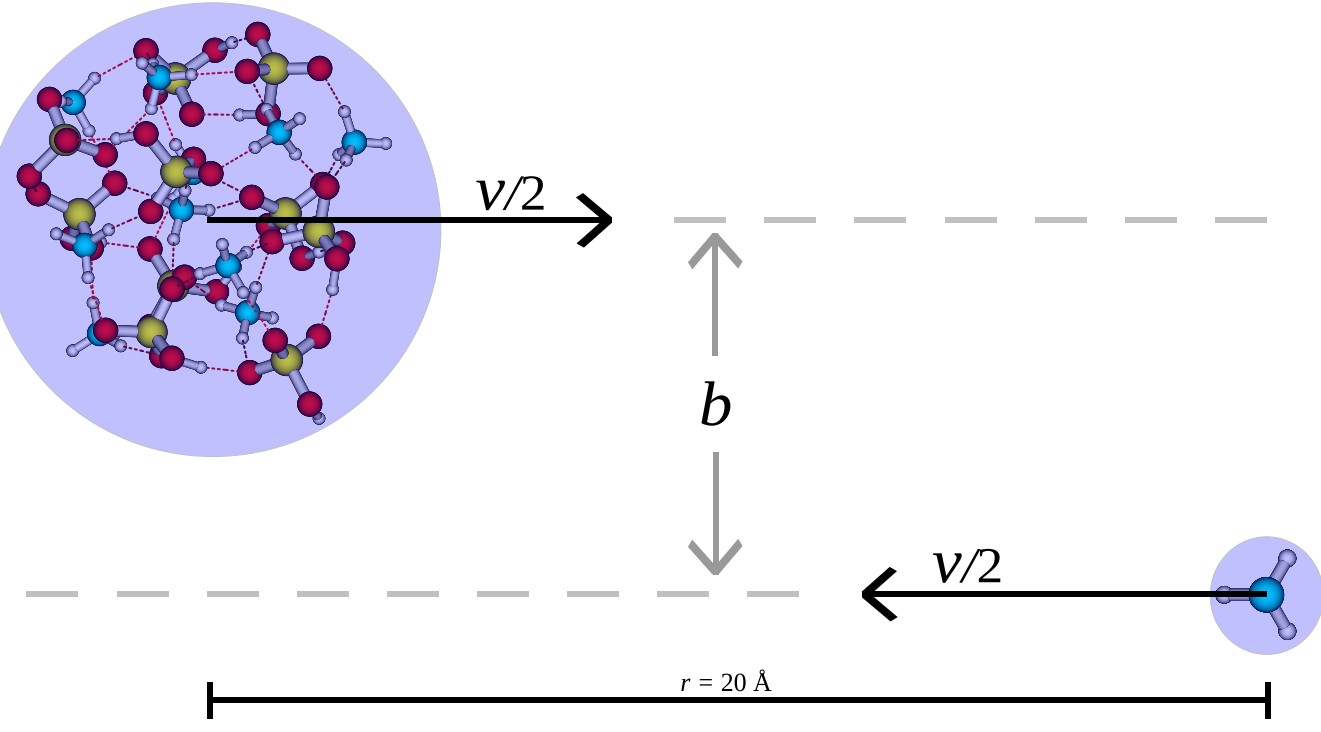

**Figure 1.** Illustration of the setup for the collision simulation. $v$ is the initial relative velocity and $b$ the perpendicular offset between the initial trajectories of the collision partners.

an intermolecular potential of the form $-A(r/r_0)^{-6}$ with $r_0 = 11.5$ Å, the sum of hard-sphere radii of the largest collision





system studied here. At 20 Å, the intermolecular potential between the collision partners drops below 5 % from the value at $r_0 = 11.5$ Å. The initial relative velocity was chosen based on the most probable velocity given by the Maxwell–Boltzmann relative velocity distribution

$$v = \sqrt{\frac{2k_{\mathrm{B}}T}{\mu}} \;\; ; \;\; \mu = \frac{m_{\mathrm{a}}m_{\mathrm{b}}}{m_{\mathrm{a}} + m_{\mathrm{b}}}, \tag{7}$$

where $\mu$ is the reduced mass of the combined system, $m_{\mathrm{a}}$ the mass of the FNP and $m_{\mathrm{b}}$ the mass of the monomer, and $T$
the temperature. It is important to establish a reasonable criterion for when a sticking collision has occurred. Since we are interested in the growth of clusters, we define the criterion based on whether the collision partners remain attached at the end of the collision simulation. The collision simulation and detection algorithm work as follows:

–  The simulation is run as long as the gradient of the COM distance is negative. Once the COM distance gradient becomes positive, we know the molecules either collided or passed each other.

–  From this point, the simulation is run for another 2 ps while storing the COM distance gradients to determine if the molecules are attached. We chose 2 ps as tests indicated that most detachments happen within the first 1 ps.

–  NO STICKING COLLISION is registered if all the gradient values are above zero, because they either bounced off each other or passed each other without interacting.

–  NO STICKING COLLISION is registered if some gradients are below zero but the final COM distance is 20 %
above the sum of hard-sphere radii of the collision partners. This means they interacted but eventually split apart.

–  Otherwise a STICKING COLLISION is registered.

We acknowledge that this sticking collision criterion is somewhat arbitrary, as the collision partners will eventually detach given enough simulation time. However, these criteria can identify attachments that remain stable for a certain period (i.e., 2 ps) after collision. While the quantitative results may vary with different criteria, we expect these variations to be small and
systematic, and therefore, do not affect the qualitative conclusions of this work.

To test this, we simulated a new head-on collision of the AM monomer with $(SA)_{10}(AM)_{10}$ at velocities of 200 and 300 m/s, tracking the collision partners 20 ps after the COM gradient change. We chose AM as we expect this to be the weakest bonding monomer. The sticking probability changed from 97 % to 98 % at 200 m/s and from 96 % to 95 % at 300 m/s. The minimal change suggests we are capturing all the rebounding events within the 2 ps window. Interestingly, the monomer already exhibits
rapid hydrogen-transfer reactions within the 2 ps simulation suggesting the monomer has begun being incorporated as part of the cluster. For the 20 ps simulations, we observe that the AM monomer has had all its hydrogen atoms exchanged several times, a process which most force field methods are unable to simulate.

## 2.4 The Sticking Coefficient

The sticking coefficient $s$ quantifies the number of sticking collisions per second per unit concentration and sets the kinetic
limit for the formation of new particles. It is typically assumed to be equal to the collision coefficient derived from kinetic gas



theory,

$$\beta_{\text{HS}} = \sqrt{\frac{8k_{\text{B}}T}{\pi\mu}} \pi \left(R_i + R_j\right)^2, \tag{8}$$

where the two collision partners are treated as non-interacting hard spheres of radius $R_i$ and $R_j$, respectively, and a collision occurs only when their radii overlap. We calculate the radius of the collision partners following the implementation in ACDC (McGrath et al., 2012). Here, the radius is given by the effective volume of a sphere which volume corresponds to the sum of the volumes of the individual monomers,

$$R = \left(\frac{3\sum_{i=1}^{N} V_i}{4\pi}\right)^{1/3}, \tag{9}$$

where the monomer volume $V_i$ for the $N$ monomers in the FNP are given by their molar mass and liquid bulk density.

The sticking coefficient can also be calculated from MD simulations via the probability of sticking $P(v,b)$ for the given offset $b$ and velocity $v$ via numerical integration of the following expression (Neefjes et al., 2022),

$$s_{\text{MD}} = \pi \int\limits_{0}^{\infty}\int\limits_{0}^{\infty} v f(v) P(b,v)\text{d}(b^2)\text{d}v, \tag{10}$$

where $f(v)$ is the Maxwell–Boltzmann relative velocity distribution for the given initial relative velocity.

The enhancement factor is then calculated as the ratio between the coefficients,

$$\eta = \frac{s_{\text{MD}}}{\beta_{\text{HS}}}. \tag{11}$$

## 3 Results and Discussion

### 3.1 Equilibration

As the equilibration of the collision partners is the most time-consuming step in our simulations, it is important to find the minimal amount of time needed to reach thermal equilibrium. The molecule with the fewest degrees of freedom is the most challenging to equilibrate. In our case, this is the AM monomer. The left and right sides of Figure 2 show the cumulative mean temperatures and temperature distributions, respectively, of the AM monomer over a 10 ns equilibration simulation. The temperature partitions are averages over the corresponding degrees of freedom. As such, their fluctuations generally decrease with the number of atoms in the system. Since AM has the fewest atoms among the studied systems, we would expect it to take the longest time to equilibrate. From Figure 2 it can be seen that the cumulative mean temperatures of each temperature partition converge around the target temperature of 300 K after $\sim$1 ns. While these cumulative mean temperatures approach the target, they do not exactly reach 300 K even after 10 ns. Specifically, the translational cumulative mean temperature achieves the target of 300 K, while the total, rotational, and vibrational cumulative mean temperatures fall short by 7 K, 1 K, and 14 K, respectively. One possible explanation for this discrepancy is that the vibrational modes are treated as quantum anharmonic oscillators at the



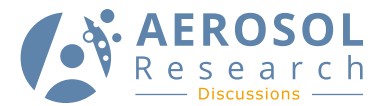

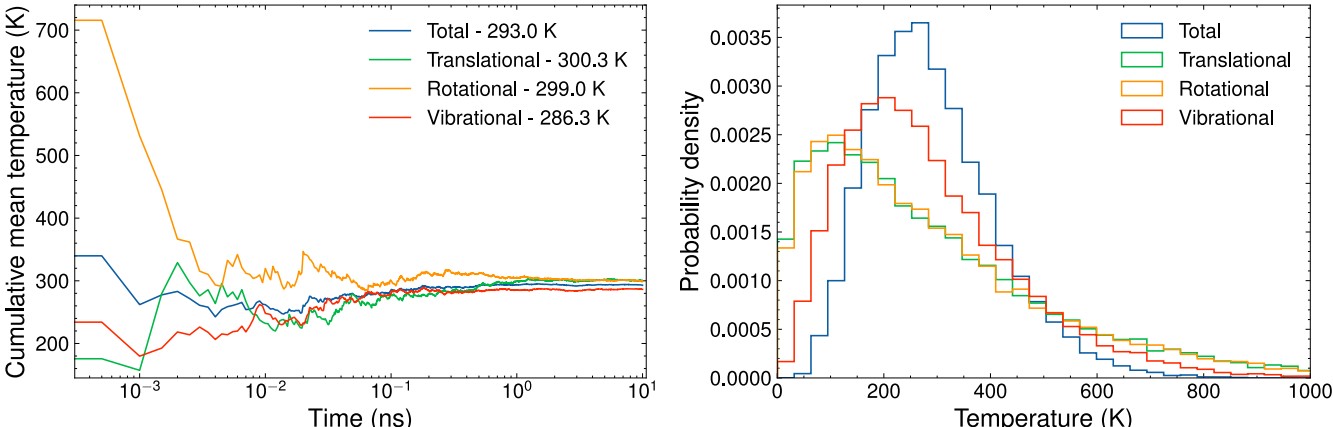

**Figure 2. (left)** Cumulative mean temperature as a function of simulation time, and **(right)** temperature distribution for a 10 ns NVT simulation of ammonia. The simulation parameters include a target temperature of 300 K, a time step of 1 fs, a Langevin thermostat time constant of 100 fs, and temperature data recorded every 500 fs.

GFN1-xTB level of theory. The equipartition theorem only strictly applies to classical degrees of freedom, where the energy is a quadratic term. Classical force field MD simulations model the bonds between atoms as classical harmonic oscillators.

Consequently, equilibration using this classical framework yields cumulative mean vibrational temperatures closer to the target temperature. However, even in classical simulations, some deviation from the target temperature occurs, likely due to the stochastic nature of the Langevin thermostat and numerical imprecision.

Although the cumulative mean temperatures do not fully converge to the target temperature during equilibration, the deviations are minor and are not expected to significantly impact the collision dynamics. The largest FNP we studied is the

$(SA)_{10}(TMA)_{10}$ cluster, consisting of 200 atoms. This cluster represents our computational bottleneck due to its size. The left and right sides of Figure 3 show the cumulative mean and distribution, respectively, of the temperature partitions for a 1 ns simulation of $(SA)_{10}(TMA)_{10}$. Given the large number of vibrational degrees of freedom, the vibrational temperature partition dominates the total temperature, and its cumulative value converges quickly toward the target temperature. In contrast, the translational and rotational temperatures, each with only three degrees of freedom, exhibit frequent fluctuations, and their

cumulative means take considerably longer to converge. Compared to AM, the larger deviations from the target temperature in the cumulative means of the translational and rotational temperature partitions are due to the greater total kinetic energy in the FNP being distributed across the same number of degrees of freedom as in the monomer. Although the cumulative mean vibrational temperature is closer to the target than for AM, it still shows a deviation of 5 K. As with AM, we consider these small deviations acceptable, as they are unlikely to significantly affect the collision dynamics. The sharp distribution of the

vibrational temperatures in Figure 3 compared to 2 is due to averaging over a larger number of atoms for the FNP.

From these results, we assume equilibration to have been achieved after 1 ns for the FNPs and 10 ns for the monomers. Following this equilibration, the next 100 output frames (saved every 500 fs) are used as input for the collision MD simula-




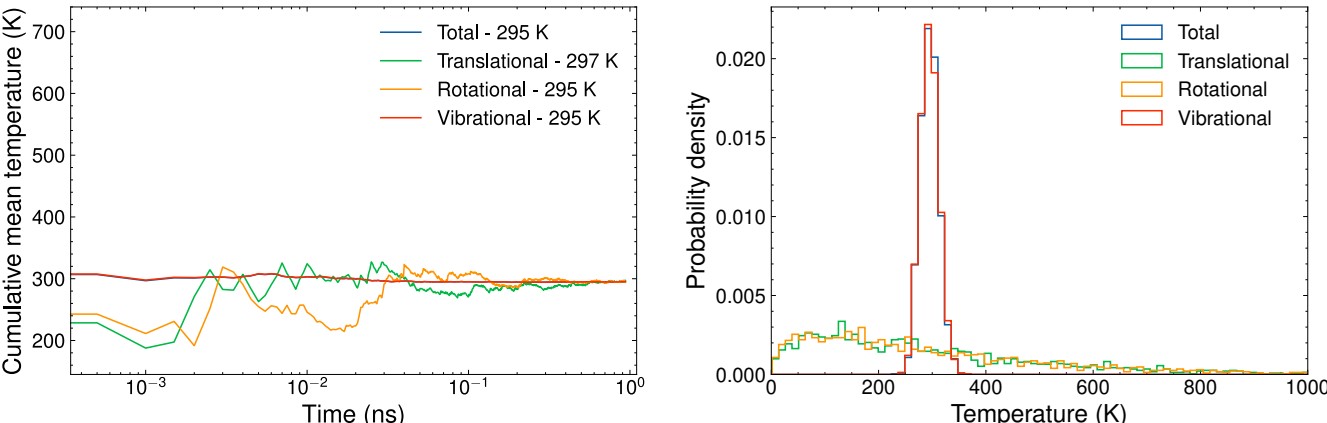

**Figure 3. (left)** Cumulative mean temperature as a function of simulation time, and **(right)** temperature distribution for a 1 ns NVT simulation of $(SA)_{10}(TMA)_{10}$. The simulation parameters include a target temperature of 300 K, a time step of 1 fs, a Langevin thermostat time constant of 100 fs, and temperature data recorded every 500 fs.

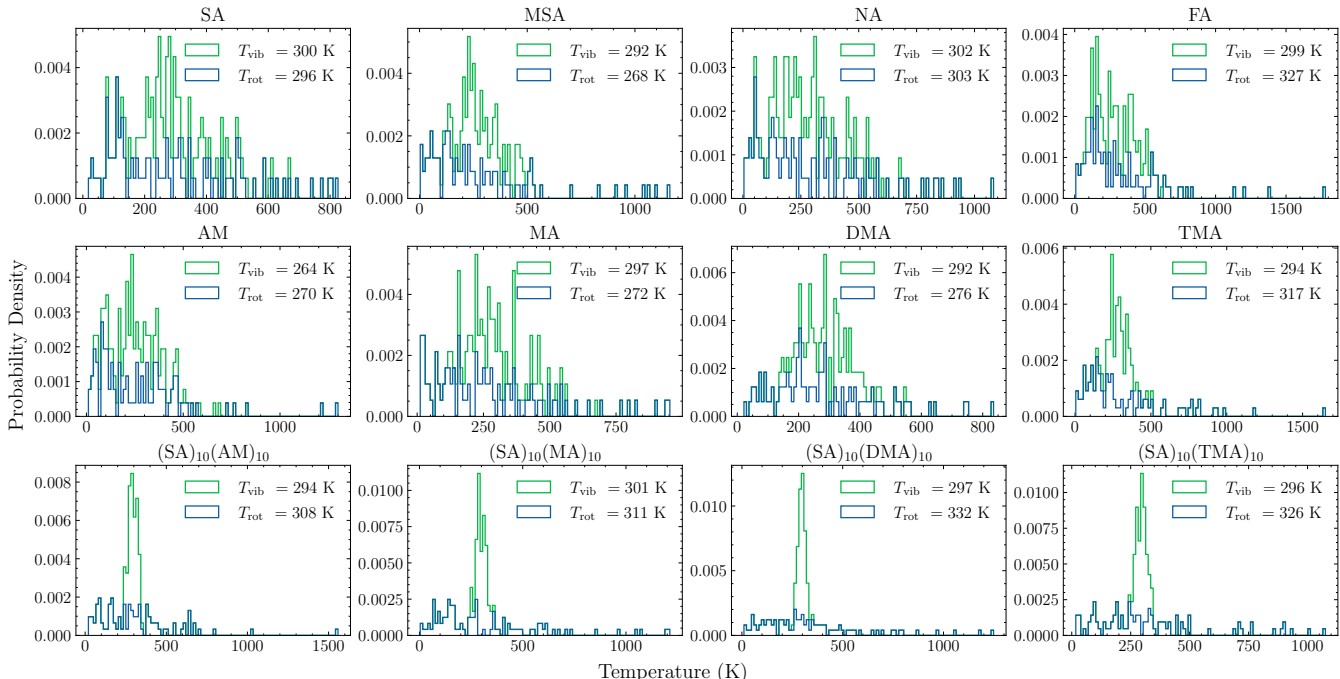

**Figure 4.** Distribution of rotational and vibrational temperature for the 100 output frames of the monomers and FNPs as obtained from the equilibration procedure.

tions. As the collision event is a non-equilibrium process for the translational degree of freedom, the center of mass is kept fixed during the equilibration and only the total, rotational, and vibrational temperatures are monitored. Figure 4 shows the





distribution of the rotational and vibrational temperatures for these 100 output frames for all studied systems, as well as their average values over the 100 output frames. The average rotational and vibrational temperatures deviate by up to 32 K from the target temperature ($T_0 = 300$). This is a result of inherent temperature fluctuations within the NVT ensemble. The expected standard deviation of the temperature is given by Landau and Lifshitz (1969),

$$\text{exp-std} = \sqrt{\frac{2}{N_{\text{dof}}}} T_0. \tag{12}$$

Overall, the standard deviations of the fluctuations align with the expected values from Equation 12, indicating that the thermostat parameters are reasonable and well-behaved. Only the vibrational temperature of AM (std $= 128$ K vs exp-std $= 173$ K) and the rotational temperatures of FA (297 K vs 245 K), DMA (169 K vs 245 K), and TMA (288 K vs 245 K) show noticeable discrepancies, possibly due to averaging over too few output frames as the sampling error is high with fewer data. These discrepancies can be reduced by including more output frames or, alternatively, collecting structures from multiple

independent equilibration runs. However, both approaches would incur a substantial increase in computational cost. Despite some deviations reaching up to 11 % from the target temperature, these variations are not expected to affect the qualitative outcomes of the collision dynamics.

### 3.2 Sticking Probabilities

Using the equilibrated structures as the starting point, we performed 100 simulations for each offset ($b$) for all the studied

systems.

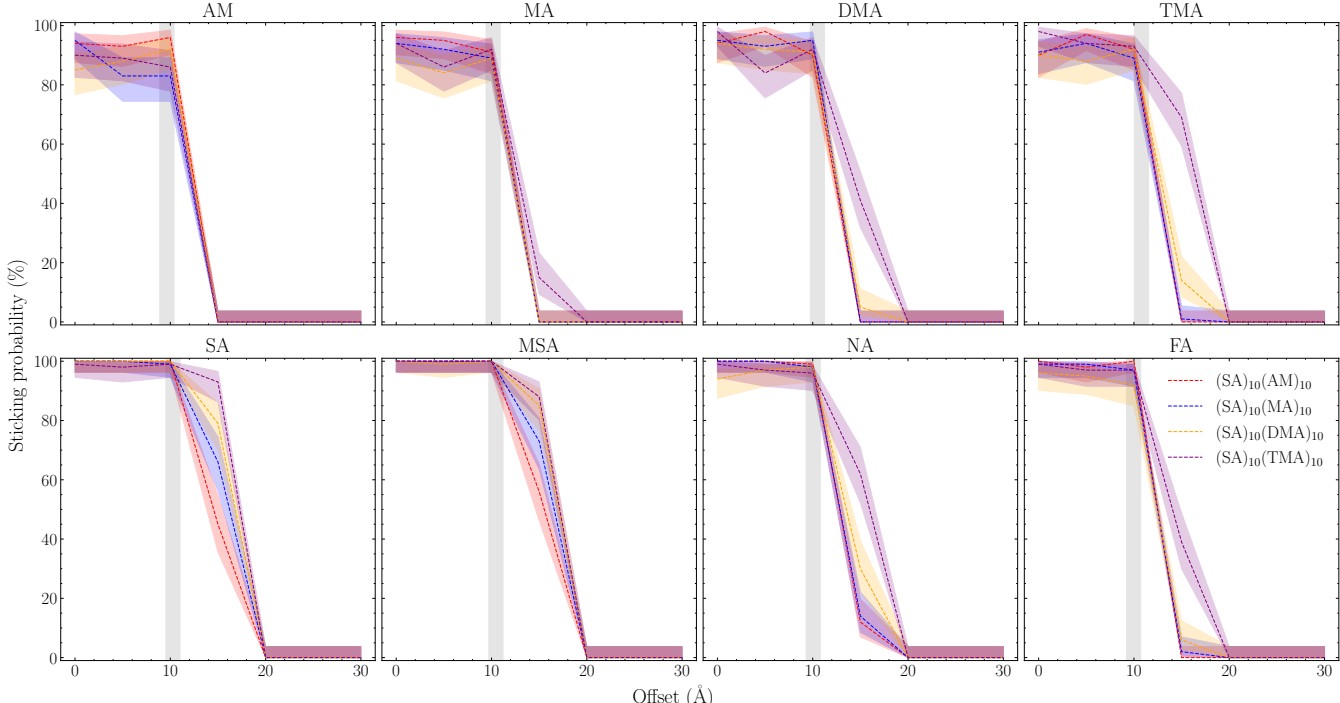





**Figure 5.** Number of collision simulations that satisfy the sticking criteria as a function of the offset between the collision partners. The uncertainty is the Wilson score interval (we assume binomial distribution). The gray area indicates the range between the smallest and largest sum of the hard-sphere radii of each system.

Figure 5 shows the sticking probabilities for the given monomer (title) onto the FNP given by the label for the given offset and at mode speed. The uncertainty is the Wilson score interval. The shaded gray area indicates the hard-sphere limits where the sticking probability from kinetic gas theory drops to 0 %. We see that for all systems, sticking occurs beyond the limits given by kinetic gas theory. This indicates that kinetic gas theory is likely underpredicting the collision coefficient as the clusters and monomers exhibit substantial long-range interactions that attract the collision partners toward each other.

For all systems, we observe a fast drop-off beyond the hard-sphere limit. This indirectly provides a measure for the strength of the long-range interactions. The strongest long-range interaction is observed for systems with the SA and MSA monomers, where it takes roughly a 10 Å offset to reach 0 % sticking. The remaining systems reach 0 % after 4–7 Å. Generally, when the individual FNPs are distinguishable (i.e., the lines are not on top of each other), we also see a drop-off that follows TMA>DMA>MA>AM for higher sticking probability. We speculate this trend is due to the size of the FNPs rather than the interaction strength, as the increasing number of methyl groups is expected to shield the partial charges, thereby reducing long-range interactions.

Interestingly, the sticking probability for the bases below the hard-sphere limit is not 100 % but instead hovers around 90 %. This suggests acid uptake on 1:1 FNPs is more favorable compared to bases, in accordance with quantum chemical calculations (Olenius et al., 2013; Elm, 2017).

If the enhanced sticking probability is due to long-range interactions, we would expect the enhancement to be proportional to the dipole moment of the monomers. To probe the correlation, we calculated the integrated area of the sticking probabilities as a function of offset for the MD simulations (Figure 5) and for kinetic gas theory (hard-sphere radii × 100 %). The ratio between the two areas should be a measure of the enhancement. The plot of the enhancement ratio versus the dipole moments of monomers and the dipole moments are shown in Figure 6.

For the $(SA)_{10}(AM)_{10}$ FNP, there is almost a direct proportionality between the ratio and the dipole moments ($R^2 = 0.963$). The $(SA)_{10}(MA)_{10}$ and $(SA)_{10}(DMA)_{10}$ are slightly less proportional with $R^2$ values of $0.900$ and $0.893$, respectively. Lastly, the $(SA)_{10}(TMA)_{10}$ FNP does not exhibit a linear proportionality between the dipoles and enhancement ratio ($R^2 = 0.494$). This is due to outliers (relative to the fit) of the TMA, MA, and FA monomers which appear to exhibit an inverse trend of enhancement ratio versus dipole moments. This shows the enhancement is quite sensitive to the constituents of the FNPs and is not entirely a function of the monomer dipole moments. We speculate that the difference is due to the shielding of the partial charges in the FNPs. For instance, the $(SA)_{10}(AM)_{10}$ FNP has no methyl groups for shielding the partial charges and it thus exhibits the direct proportionality, while the $(SA)_{10}(TMA)_{10}$ FNP has three methyl groups per base monomer and does not exhibit a strong proportionality.

Interestingly, the $R^2$ values for $(SA)_{10}(MA)_{10}$ and $(SA)_{10}(DMA)_{10}$ are remarkably similar to those of $(SA)_{10}(AM)_{10}$, despite the presence of additional methyl groups. We speculate that the influence of the additional methyl groups is counteracted by the accompanying increase in dipole moment, resulting in little to no change.





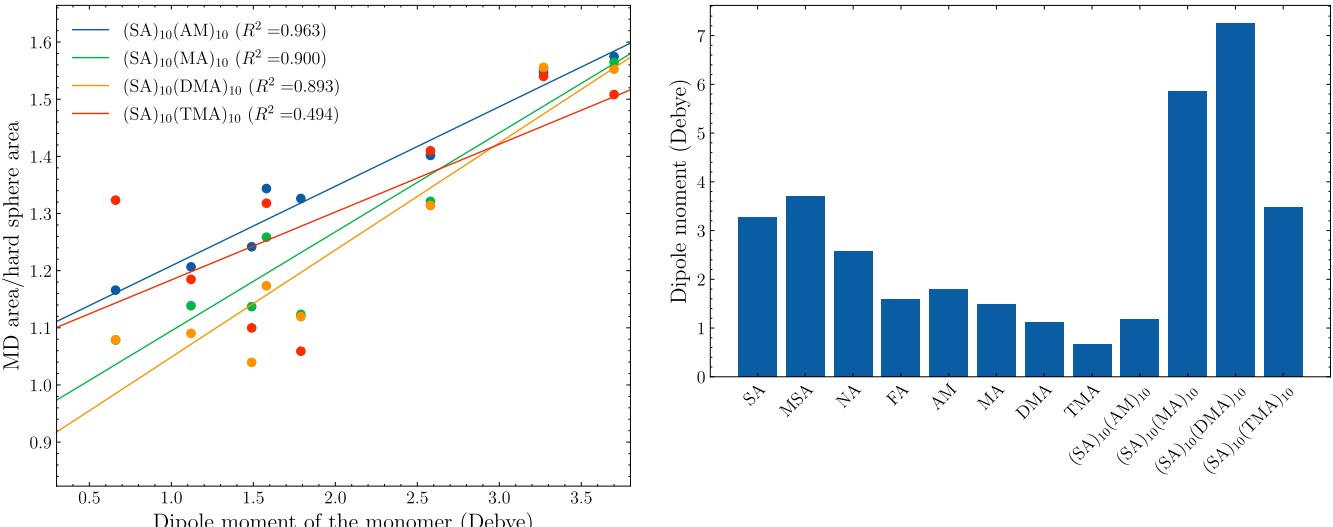

**Figure 6. (left)** The ratio between the area of the MD simulation in Figure 5 and the area of probability calculated by kinetic gas theory as a function of the dipole moment of the monomer. **(right)** The dipole moment of the monomers calculated at the $\omega$B97X-D/6-31++G(d,p) level of theory. The FNPs are at the B97-3c level of theory.

### 3.2.1 Sticking Coefficient

To obtain an explicit measure of the enhancement, the sticking coefficient must be calculated. This requires simulating sticking collisions for all initial relative velocities that are significantly populated according to the Maxwell–Boltzmann relative velocity distribution. This is computationally expensive for the larger FNPs, such as $(SA)_{10}(TMA)_{10}$, as the low-velocity simulations would take a long time at the GFN1-xTB level. However, it is feasible for the smaller $(SA)_{10}(AM)_{10}$ FNP, where we simulate the SA and AM collisions for velocities of 50 and 100–1300 ms$^{-1}$ in increments of 100, and an offset of 0 to 30 Å in steps of

2 Å. From Figure 7, we again observe sticking probabilities above the limit given by the sum of hard-sphere radii (green line). This is especially noticeable for the sulfuric acid collision, where the most populated velocities (100–400 m/s) have significant sticking probabilities for offsets up to 6 Å larger than the hard-sphere limit. For the SA collision, the assumption of 100 % sticking probability below the hard-sphere limit is valid for all the populated initial relative velocities. For the AM collision, this assumption does not hold, as several points below the limit have probabilities around 80 %. However, it still exhibits

sticking probability beyond the hard-sphere limit, but it only extends for 2 Å at the most populated velocities (400–700 m/s). AM being less likely to form stable clusters compared to SA is consistent with quantum chemical calculations, which indicates that cluster growth tends to favor acid-first formation followed by base addition(Olenius et al., 2013; Elm, 2017; Kubečka et al., 2023).

Following the edge along the exponential decreasing curve, the systems exhibit a sharp drop to zero probability. This indi-

cates the interactions and uptake process behave isotropically. If the sticking probability is strongly orientation-dependent, we



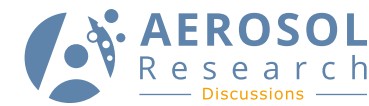

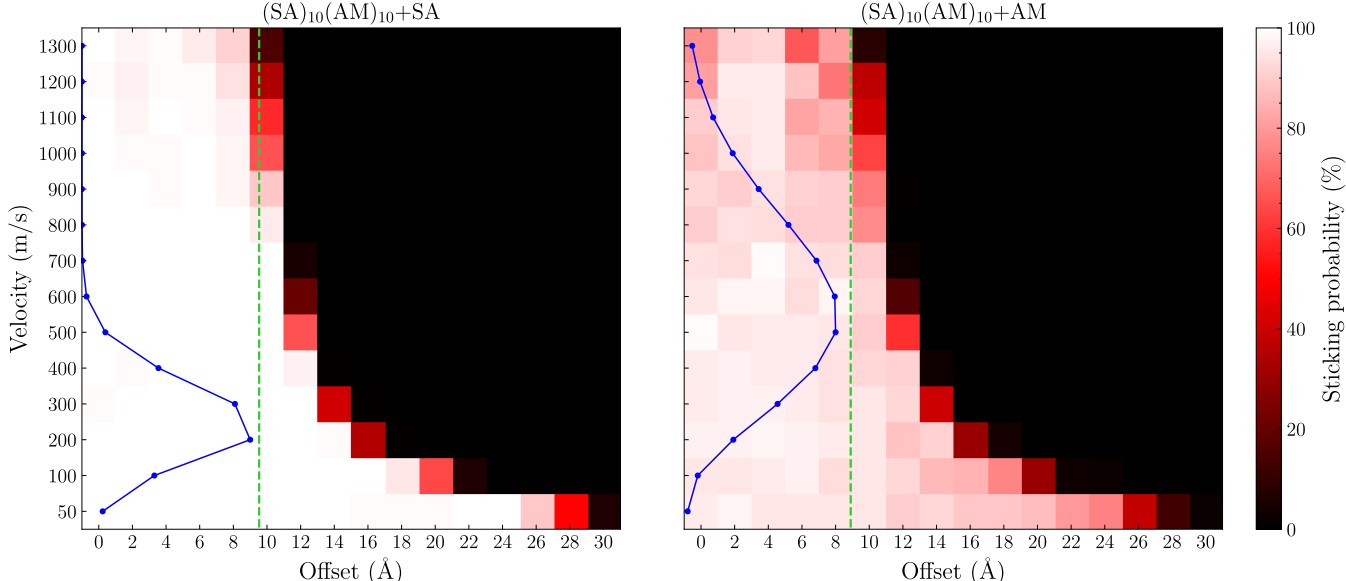

**Figure 7.** Grid of the sticking probability for the **(left)** SA+$(SA)_{10}(AM)_{10}$, **(right)** AM+$(SA)_{10}(AM)_{10}$ collision as a function of the offset and velocity. Each point was simulated 100 times using GFN1-xTB. The blue points show the Maxwell–Boltzmann distribution at the relative velocity. The green line indicates the sum of the hard-sphere radii of the system.

would expect a more gradual drop-off as a function of offset as a subset of the possible orientations would lead to sticking. It should be noted that this is less pronounced for the AM collision, as it shows more variation in the sticking probabilities as a function of the offset, especially for the lower velocities (50–200 m/s). However, these are mostly in the 80–99 % range, and are therefore still not strongly orientation-dependent.

Integrating the attachment probability using Equation (10), we find an enhancement factor $\eta$ of 2.3 and 1.5 for the SA+$(SA)_{10}(AM)_{10}$ and AM+$(SA)_{10}(AM)_{10}$ collisions, respectively. The SA enhancement factor also matches the one found by Yang et al. (2023) (roughly 2.5) for the SA+$(SA)_n(DMA)_n$ collision using the OPLS all-atom force field at similar sizes. It should be noted that these enhancement factors are quite sensitive to the grid sizes simulated and the chosen numerical integration method. Therefore, rounding to the nearest half-point is reasonable given the uncertainty. For both systems, we underestimate the collision

coefficient when using the hard-sphere model. The enhancement scales with the interaction strength, as the stronger interacting SA molecule, has an enhancement of roughly 2, while the weaker interacting AM molecules only exhibit an enhancement of 1.5. While not all AM collisions below the hard-sphere limit have a sticking probability of 100 %, the increased probability from the ranged interactions still results in an enhancement factor above 1.

     If all collisions in the ACDC simulation include an enhancement factor of 2, the simulated new particle formation rate

doubles. This indicates that the standard ACDC settings may underestimate the actual new particle formation rate.





### 3.2.2 Comparing with Force Field Methods

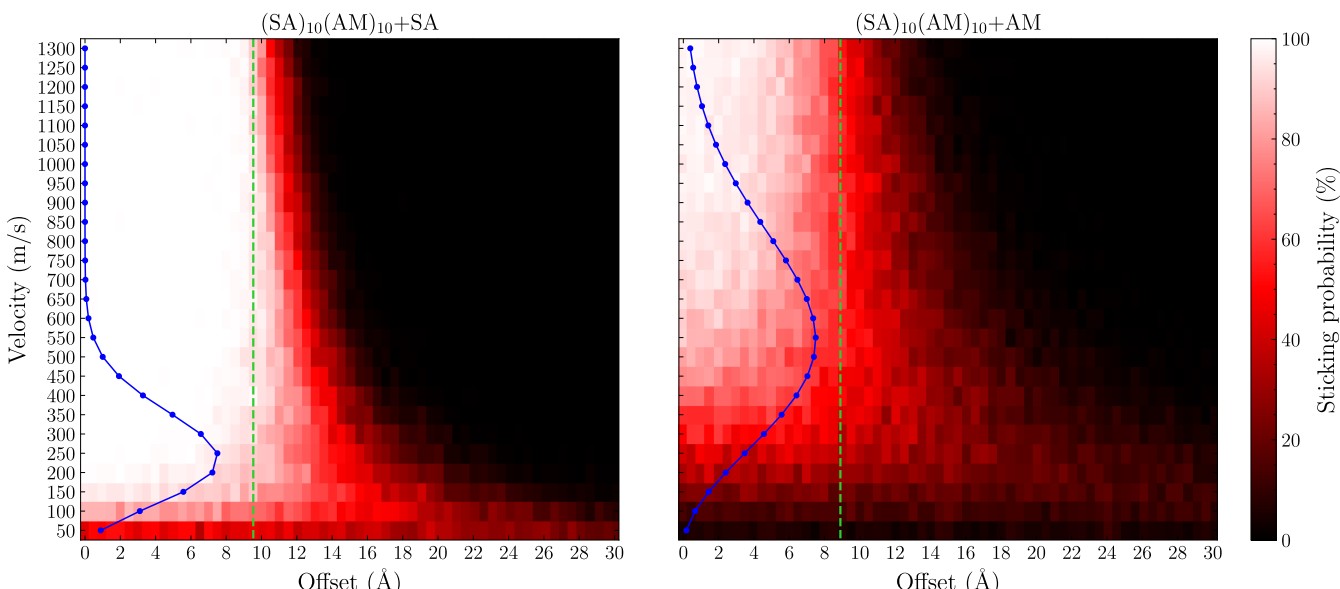

**Figure 8.** Grid of the sticking probability for the **(left)** SA+(SA)$_{10}$(AM)$_{10}$, **(right)** AM+(SA)$_{10}$(AM)$_{10}$ collision as a function of the offset and velocity. Each point was simulated 200 times using the OPLS force field. The blue points show the Maxwell-Boltzmann distribution at the relative velocity. The green line indicates the sum of the hard-sphere radii of the system.

To compare the sticking probabilities from the semi-empirical method with those from force field calculations, we performed simulations for the same system using the methodology of Yang et al. (2023), employing the OPLS all-atom force field .

We find the force field enhancement factors $\eta$ to be 2.4 and 1.6 for the SA+(SA)$_{10}$(AM)$_{10}$ and AM+(SA)$_{10}$(AM)$_{10}$ col-
lisions, respectively. While the enhancement factors are similar to the GFN1-xTB simulations, the sticking behavior is quite different. The force field method exhibits a more gradual decrease in the probability as a function of offset, suggesting a more anisotropic behavior compared to GFN1-xTB. For lower velocities, sub 200 m/s for SA and sub 400 m/s for AM, we observe sticking probability at or below 50 % almost independently of the offset. This has previously been attributed to periodic repulsions between the rotating collision partners due to the orientation of their fixed charge distributions (Halonen et al., 2019).
However, this behavior does not occur in the GFN1-xTB calculations. The electron distribution modeled by the semi-empirical method is able to accommodate the relative orientations between the collision partners during their flight resulting in the isotropic behavior. This discrepancy does not lead to substantial differences in the enhancement factor, as only a small fraction of the total collision system population occupies these low relative velocities. However, we would expect the enhancement factor of the two methods to diverge for larger effective masses, where the lower velocities are more significantly populated.
Likewise, we would also expect the two methodologies to differ for sticking events where reactions are extremely important. For instance, hydrogen transfer reactions for cluster and monomer collision that would form a cluster with 1:1 acid–base ratio





will make the monomer stick stronger. However, force field simulations would not capture this, as most force field methods are unable to simulate bond breaking and formation.

## 4    Conclusions

We investigated the sticking process of the atmospherically-relevant monomers sulfuric acid (SA), methanesulfonic acid (MSA), nitric acid (NA), formic acid (FA), ammonia (AM), methylamine (MA), dimethylamine (DMA), and trimethylamine (TMA) onto freshly nucleated particles (FNPs). The FNPs considered are $(SA)_{10}(AM)_{10}$, $(SA)_{10}(MA)_{10}$, $(SA)_{10}(DMA)_{10}$, and $(SA)_{10}(TMA)_{10}$. The simulations were performed using semi-empirical molecular dynamics (SEMD) at the GFN1-xTB level of theory, which, unlike classical force field methods, accounts for chemical reactions and the dynamic charge distribution.

Using the equipartition theorem as a measure for thermal equilibrium, we find that the collision partners require an equilibration run of 10 ns for the monomers and 1 ns for the FNPs. Equilibration was performed using the Langevin thermostat targeting 300 K, with a thermostat time constant of 100 fs, and a time step of 1 fs.

Carrying out the simulations using the equilibrated collision partners, we find that the hard-sphere kinetic approximation, which neglects long-range interactions, significantly underestimates the number of collisions leading to sticking. Comparing the acids and bases, the acids show a higher probability due to the increased long-range interaction strength. Furthermore, the sticking probability for the bases below the hard-sphere limit is not 100 % but instead hovers around 90 %.

We find enhancement factors of 2.3 and 1.5 for the SA+$(SA)_{10}(AM)_{10}$ and AM+$(SA)_{10}(AM)_{10}$ again confirming kinetic gas theory underestimates the sticking probability. Comparing the enhancement factors with those obtained from OPLS allatom force field simulations, we find similar values of 2.4 and 1.6 for SA+$(SA)_{10}(AM)_{10}$ and AM+$(SA)_{10}(AM)_{10}$ collision, respectively. Although the enhancement factors are similar there are two major differences: 1) At lower collision velocities—below 200 m/s for SA and below 400 m/s for AM—force field simulations predict low or zero sticking probability, even for head-on collisions. 2) In the semi-empirical simulations, the sticking probability remains near unity for small offsets and then drops sharply to 0% beyond a certain offset, exhibiting an almost isotropic behavior. In contrast, the force field simulations show a more gradual decline in sticking probability, suggesting anisotropic behavior, where certain orientations still lead to sticking while others do not.

The first point accounts for the largest difference in sticking probabilities, but since these velocities are not significantly populated in our simulations, both methods yield similar enhancement factors. However, for systems with larger effective masses, we expect the enhancement factors from force field and semi-empirical methods to diverge Additionally, for sticking events where hydrogen transfer reactions are important (e.g., when going from a non-diagonal cluster to a cluster with 1:1 acid–base ratio), the semi-empirical method will likely predict higher sticking probabilities.

This study demonstrates the use of semi-empirical methods in simulating particle formation dynamics. These methods overcome the limitations of classical force field approaches, while remaining efficient enough to enable a sufficient number of simulations for accurately calculating sticking coefficients. This represents a step toward more accurate simulations of atmospheric particle formation and growth processes.



*Data availability.*


*Author contributions.* Conceptualization: J.E.;

Methodology: Y.K., I.N., J.K., J.E.;

Formal analysis: Y.K. I.N., J.K.;

Investigation: Y.K. I.N., J.K.,;

Resources: J.E.;

Writing - original draft: Y.K., I.N.;

Writing - review & editing: Y.K., I.N., J.K., J.E.;

Visualization: Y.K.;

Project administration: J.E.;

Funding acquisition: J.E;

Supervision: J.E.

*Competing interests.* At least one of the (co-)authors is a member of the editorial board of Aerosol Research. The authors have no other competing interests to declare.

*Acknowledgements.* Funded by the European Union (ERC, ExploreFNP, project 101040353 and MSCA, HYDRO-CLUSTER, project
101105506). Views and opinions expressed are however those of the authors only and do not necessarily reflect those of the European Union or the European Research Council Executive Agency. Neither the European Union nor the granting authority can be held responsible for them.

This work was funded by the Danish National Research Foundation (DNRF172) through the Center of Excellence for Chemistry of Clouds.

The numerical results presented in this work were obtained at the Centre for Scientific Computing, Aarhus https://phys.au.dk/forskning/ faciliteter/cscaa/.



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
