# Peer review of "Growth of Atmospheric Freshly Nucleated Particles: A Semi-Empirical Molecular Dynamics Study"

_Aerosol Research, 2025_

## Author Response (AR1)

**Response to Reviews**

We highly appreciate the positive comments from both reviewers. All points raised have been addressed in the revised manuscript. We hope that the responses below meet the reviewers' expectations and that the manuscript can be accepted for publication. The reviewers' comments have been reproduced in blue text below, followed by our point-by-point replies.

**Anonymous Referee #1**

The authors studied the collision and sticking processes of multiple acid and base molecules onto acid-base clusters with 10 acid-base pairs, using the semi-empirical GFN1-xTB method. The results were compared with those from the classical force field methods. The results are of interest, especially most previous studies on atmospheric cluster collision and sticking were performed using non-reactive force fields, while very few report results with a method that is reactive. The study is comprehensive and well carried out. I support its publication in Aerosol Research, with only a few minor comments:

Many readers might be interested to see more discussions/details on how proton transfer reactions happen at the molecule-cluster interface upon collision, and how that could vary the evaporation tendency of the newly captured molecule. I would encourage the authors to provide more details on this point. Maybe some statistics and visualizations could be helpful.

**Author reply:**

We agree that such a discussion would be beneficial to the manuscript. To address this, we calculated the distance from the nitrogen in the ammonia collision partner to the nearest hydrogen which does not originate from the incoming ammonia monomer. This allowed us to quantify the frequency of proton transfer. We have added a paragraph about this to the manuscript.

**Line 173, Page 7**

**From:**

Interestingly, the monomer already exhibits rapid hydrogen-transfer reactions within the 2 ps simulation suggesting the monomer has begun being incorporated as part of the cluster. For the 20 ps simulations, we observe that the AM monomer has had all its hydrogen atoms exchanged several times, a process which most force field methods are unable to simulate.

To:

Interestingly, if the monomer is aligned with a hydrogen on a sulfuric acid for the initial collision, the monomer exhibits rapid hydrogen-transfer reactions within the 2 ps simulation. This can be seen from the rapid initial drop in Figure 1, which shows the distance to the nearest hydrogen atom not originally attached to the monomer as a function of simulation time after the collision criterion has been fulfilled. Likewise, we also observe rapid back-and-forth hydrogen transfers later in the simulation when the nearest hydrogen gets aligned with an oxygen from sulfuric acid. This behavior is exhibited as large oscillating values in the nearest-distance plot (see 2000–4500 fs in Figure 1).

Figure 1: The distance from the nitrogen in the ammonia collision partner to the nearest hydrogen that does not originally come from the incoming ammonia monomer. The data is from one of the 300 m/s collision simulations of AM and  $(SA)_{10}(AM)_{10}$  from the point where a collision is detected.

Here, we define a hydrogen transfer to occur when the distance between nitrogen and a new hydrogen drops below 1.1 Å. Accordingly, we find that the transfer occurs, for 63 % (21 % with the first two ps) and 58 % (11 % within the first two ps) for the 300 m/s and 200 m/s simulations, respectively. The fast hydrogen transfer allows the monomer to be more easily incorporated into the cluster. Nevertheless, the redistribution of excess energy still plays a crucial role, as a sudden excess of translational energy on any cluster molecule can lead to the rapid loss of hydrogen and evaporation.

Line 74: To my knowledge, Yang et al (Yang H, Goudeli E, Hogan C J. Condensation and dissociation rates for gas phase metal clusters from molecular dynamics trajectory calculations[J]. The Journal of chemical physics, 2018, 148(16).) employed the same simulation setup and method to calculate the sticking rates. It is also earlier than the other papers, which used the same method, cited here. It might not be bad to also cite this paper.

**Author reply:**

We appreciate the supplied reference and agree that including the suggested paper in the introduction would be beneficial.

**Line 83, Page 2**

From:

Halonen et al. used force field molecular dynamics to study collisions between two SA molecules, finding a collision coefficient 2.2 times larger than that predicted by kinetic gas theory.

To:

Yang et al. used a force field molecular dynamics setup to study collisions of Au/Mg atoms and clusters and quantify the collision coefficient. Halonen et al. used a similar setup to study the collision between two SA molecules, finding a collision coefficient 2.2 times larger than that predicted by kinetic gas theory.

Line 134: The form of the intermolecular potential might not be a good approximation for molecule-cluster interactions. Maybe the authors could consider to check the acceleration of the molecule instead, in order to make sure if the initial COM distance is large enough?

**Author reply:**

We agree that it is not necessarily clear that the potential will follow this form for our clusters. To obtain a general estimate of the interaction strength as a function of center-of-mass distance, we calculated the potential of mean force between 10sa10am and 1sa through umbrella sampling using the OPLS all-atom force field to probe if we are out of the potential well. This discussion has been added to the support information as well as a reference to it in the main manuscript.

**Added in SI:**

To obtain a general estimate of the interaction strength as a function of center-of-mass distance, we calculated the potential of mean force between  $(SA)_{10}(AM)_{10}$  and SA through umbrella sampling (Torrie and Valleau, 1977) using the OPLS all-atom force field (Jorgensen et al., 1996). The umbrella sampling simulations were performed in LAMMPS (Large-scale Atomic/Molecular Massively Parallel Simulator (Plimpton, 1995; Thompson et al., 2022)) using the PLUMED plug-in (Tribello et al., 2014). We sampled center-of-mass distances from 6.0 to 40.0 Å in windows of 0.4 Å.

Figure 2: The potential of mean force for the  $(SA)_{10}(AM)_{10}$ -SA system using OPLS-all atom force field.

In each window, the  $(SA)_{10}(AM)_{10}$ –SA system was placed at the corresponding distance and equilibrated for 1 ns using a Langevin thermostat (Schneider and Stoll, 1978; Bussi and Parrinello, 2007) at 300 K with a bias potential of 4 eV/Å, followed by a 100 ns production run employing a canonical sampling through velocity rescaling thermostat (Bussi et al., 2007) and a bias potential of 0.5 eV/Å.

The timestep of the simulations was set to 1 fs. Following the simulations, the free energy profile was constructed using the WHAM code (Grossfield, 2002). The PMF was then corrected by adding the term  $+k_{\rm B}T\ln r^2$  to account for configurational entropy. Force field parameters for OPLS were taken from Loukonen et al. (2010).

**Line 143, Page 5**

From:

The placement distance is motivated by assuming an intermolecular potential of the form  $-A(r/r_0)^{-6}$  with  $r_0 = 11.5$  Å, the sum of hard-sphere radii of the largest collision system studied here. At 20 Å, the intermolecular potential between the collision partners drops below 5 % from the value at  $r_0 = 11.5$  Å.

To:

The motivation for the placement distance is as follows: To determine a reasonable initial distance between the collision partners, where interactions have negligible effect on the collisions, we calculated the potential of mean force between  $(SA)_{10}(AM)_{10}$  cluster and SA monomer along the center-of-mass distance through umbrella sampling (Torrie and Valleau, 1977) using the OPLS all-atom force field (Jorgensen et al., 1996) (see section S1). The potential of mean force approaches zero around 15 Å. While this provides a good indication of the interaction strength, the GFN1-xTB method may predict slightly stronger interactions. We therefore set the initial distance to 20 Å as the overall interaction force applied over time at and beyond this point should have a negligible effect on the collision probability.

**Mária Lbadaoui-Darvas Referee #2**

The authors present a comprehensive study of collisional growth of freshly nucleated particles using semi-empirical molecular dynamic at the GFN2-xTB level and characterise the new method as an alternative to hard sphere model approximations or classical molecular dynamics simulations. They investigate the uptake (sticking) of acidic and basic molecules on a freshly nucleated particle consisting of 10 sulfuric acid and 10 trimethylamine molecules. They estimate collision enhancement factors and sticking probabilities and compare the newly presented method with standard forcefield simulations as well with calculations based on the kinetic gas theory.

They conclude that the kinetic gas theory underestimates the sticking probability and find similar results as using classical molecular dynamics simulation. They highligh that the two latter methods might diverge for particles with larger masses and in case of proton transfer reactions which are neglected in standard forcefield simulations. The method presented in the manuscript has clear advantages over traditional methods used to estimate sticking probabilities which are essential for performing cluster dynamics models.

The publication is well written, and results represent an important contribution to the field, therefore I warmly recommend the paper to be published after the following minor comments are addressed:

The results and the context are very well presented and explained in details, however the description of the molecular simulations used to produce the data could be extended. In the current format many technical concepts, such as SEMD and the level of theory used in the calculation, remain only fully understandable to experts of MD simulations, while the journal targets a more general audience. Keeping this latter consideration in mind, I think that the paper would benefit from adding short description of SEMD and highlighting main differences with classical MD.

**Author reply:**

We agree that a description SEMD and its differences between classical MD would be beneficial for the manuscript.

**Added: Line 74, Page 3**

The choice of the potential energy surface (PES) on which the nuclei move determines the character of an MD simulation. In classical MD, the PES is parameterized based on classical mechanics terms of intra- and intermolecular interactions. These functions are relatively simple, allowing for simulations of large molecular systems, but bonds are usually based on a harmonic potential and therefore cannot model bond breaking.

By contrast, in Born–Oppenheimer MD (BOMD), the forces (the derivatives of the PES)

are obtained directly from electronic-structure calculations by solving the time-independent Schrödinger equation. In principle, BOMD can employ any level of theory to compute energies and forces, but this rapidly becomes prohibitively expensive for large systems. Semi-empirical methods strike a practical compromise by replacing the most computationally intensive components with parametrized, empirical expressions. While these methods introduce approximations, they remain rooted in an electronic-structure framework and thus retain the ability to simulate bond breaking and formation.

I would appreciate a comparison of computational cost of the SEMD and the corresponding classical MD simulations.

**Author reply:**

We agree and have included the following paragraph to address it.

**Added: Line 204, Page 8**

As an example, it takes 14 minutes for 1000 steps in a velocity Verlet simulation of the  $(SA)_{10}(AM)_{10}$  cluster using 1 CPU at GFN1-xTB. The OPLS force-field takes 0.1 seconds for the same setup, i.e., 3 to 4 orders faster.

Line 65 (Furthermore, it is not clear if the unit accommodation factor is a reasonable approximation for all clusters of interest in NPF) would fit better as the first sentence of the following paragraph

**Author reply:**

We are unfortunately unclear about where the reviewer intends for us to move this sentence. The paragraph that immediately follows discusses experimental measurements and is unrelated to the underlying assumptions involved in modeling via kinetic gas theory.

If "following paragraph" refers to a specific paragraph that got lost during the review process: The "unit accommodation factor" refers to the properties of the sticking coefficient being calculated with kinetic gas theory, which is introduced in the previous sentence. As such, relocating the sentence earlier in the manuscript would require substantial restructuring of the introduction.

In Figure 3, the total cumulative mean temperature is hardly visible, it would be perhaps meaningful to use a different linestyle that allows overlapping lines to show up.

**Author reply:**

We agree that the total cumulative mean temperature could be more visible. We have increased the linewidth to make it visible under the vibrational temperature.

**Updated Figure 3**